# Autophagy Mediates *Escherichia Coli*-Induced Cellular Inflammatory Injury by Regulating Calcium Mobilization, Mitochondrial Dysfunction, and Endoplasmic Reticulum Stress

**DOI:** 10.3390/ijms232214174

**Published:** 2022-11-16

**Authors:** Jianguo Liu, Rendong Qiu, Ran Liu, Pengjie Song, Pengfei Lin, Huatao Chen, Dong Zhou, Aihua Wang, Yaping Jin

**Affiliations:** Key Laboratory of Animal Biotechnology of the Ministry of Agriculture, College of Veterinary Medicine, Northwest A&F University, Yangling, Xianyang 712100, China

**Keywords:** autophagy, inflammation, calcium mobilization, mitochondrial dysfunction, endoplasmic reticulum stress, endometritis

## Abstract

Bovine endometritis is a reproductive disorder that is induced by mucus or purulent inflammation of the uterine mucosa. However, the intracellular control chain during inflammatory injury remains unclear. In the present study, we found that *E. coli* activated the inflammatory response through the assembly of the NLRP3 inflammasome and activation of the NF-κB p65 subunit in primary bovine endometrial epithelial cells (bEECs). Infection with *E. coli* also led to an abnormal increase in cytoplasmic calcium and mitochondrial dysfunction. Additionally, live-cell imaging of calcium reporters indicated that the increase in cytosolic calcium mainly was caused by the release of Ca^2+^ ions stored in the ER and mitochondria, which was independent of extracellular calcium. Cytoplasmic calcium regulates mitochondrial respiratory chain transmission, DNA replication, and biogenesis. Pretreatment with NAC, BAPTA-AM, or 2-APB reduced the expression of IL-1β and IL-18. Moreover, ERS was involved in the regulation of bovine endometritis and cytosolic calcium was an important factor for regulating ERS in *E. coli*-induced inflammation. Finally, activation of autophagy inhibited the release of IL-1β and IL-18, cytochrome c, ATP, ERS-related proteins, and cytoplasmic calcium. Collectively, our findings demonstrate that autophagy mediated *E. coli*-induced cellular inflammatory injury by regulating cytoplasmic calcium, mitochondrial dysfunction, and ERS.

## 1. Introduction

Bovine endometritis is a common reproductive disorder in clinical practice and frequently results in endometrial structural damage and bovine infertility, thereby causing significant economic loss to the bovine breeding industry [1]. Postpartum endometrial bacterial infection in bovine is a highly dynamic process in which the infectious species follow a characteristic progression pattern. An imbalance in the vaginal flora is the fundamental factor that leads to the high incidence and significant hazards that are associated with bovine endometritis [2]. Based on the isolation, culture, and classification of postpartum uterus bacteria, diverse species mediate infection, including *Escherichia coli*, *Trueperella pyogenes*, *Fusobacterium necrophorum*, *Prevotella melaninogenica*, and *Proteus* spp., although *E. coli* and *T. pyogenes* are the most common uterine pathogens [3,4,5]. The NOD-like receptor pyrin domain containing 3 (NLRP3) is an intracellular protein complex that assembles in response to certain pathogenic microorganisms or sterile danger signals and induces an inflammatory response. The NLRP3 inflammasome consists of NLRP3, ASC, and Caspase-1. The inactive precursors IL-1β and IL-18 are cleaved into biologically active IL-1β and IL-18, respectively, when the inflammasome is activated [6,7]. Additionally, binding of pathogen-associated molecular patterns (PAMPs) to Toll-like receptors (TLR) activates the NF-κB signaling pathway, which results in the production of inflammatory factors, including IL-1β, IL-6, and IL-8 [8,9]. Furthermore, excessive secretion of inflammatory factors in endometrial epithelial cells (EECs) leads to endometrial tissue damage, which promotes cell cytotoxicity [10].

Calcium ions are crucial for regulating numerous cellular processes, including metabolism, proliferation, differentiation, and gene transcription [11]. Analogously, mitochondria are ubiquitous, multifunctional, and dynamic and determine the regulation of multiple cellular functions. The roles of calcium and mitochondria are deeply interconnected: mitochondria regulate Ca^2+^ ion dynamics and Ca^2+^ ions modulate mitochondrial functions [12]. Abnormal calcium signaling may lead to mitochondrial dysfunction, cell damage, and ultimately cell death [13]. Additionally, the endoplasmic reticulum (ER) maintains cellular homeostasis by controlling intracellular levels of both free and bound calcium ions. When various pathological factors lead to the disturbance of calcium signaling, ER stress (ERS) is activated and coping responses, including the unfolded protein response (UPR), are mobilized to restore ER homeostasis [11]. A growing number of studies have shown that cytosolic Ca^2+^ levels, mitochondrial activity, and ERS regulate the development and outcome of inflammatory responses. For example, abnormal increases in cytoplasmic Ca^2+^ ions, mitochondrial dysfunction, and ERS are associated with acute pancreatitis in mice and rats [14]. However, the precise regulatory mechanisms that interconnect Ca^2+^ ion mobilization, mitochondrial dysfunction, and ERS in endometritis are unclear. Moreover, the roles of Ca^2+^ ions in the regulation of mitochondrial oxidative damage and during ERS in bovine endometritis are uncertain.

Autophagic processes exert quality control over cellular components, regulate metabolism, and enhance innate immunity [15]. Normative autophagy regulates biological processes in multiple cell types and promotes cellular homeostasis and survival [16]. Perturbation of autophagy impacts a variety of diseases, including microbial infections, autoimmunity, metabolic disorders, and neurodegenerative diseases [17,18]. Conversely, autophagy as a cytoplasmic degradation pathway has protective effects against a variety of exogenous and endogenous hazards [19]. This study focused on the role of autophagy in bovine endometritis, including the secretion of inflammatory factors and export of inflammasomes, with particular emphasis on the intersection of autophagy with innate immune responses, second messengers, and organelle functions, including mitochondria and ER, which are the platforms of immune metabolic signaling. The results reveal that autophagy mediated *E. coli*-induced cellular inflammatory injury by regulating cytoplasmic calcium levels, mitochondrial dysfunction, and ERS.

## 2. Results

### 2.1. E. Coli Activated Inflammatory Response in bEECs

We measured the levels of pre-IL-1β and pre-IL-18 cytokine proteins in primary bEECs to assess the role of *E. coli* in stimulating an inflammatory response (Figure 1A). The levels of these cytokines increased significantly after *E. coli* infection. Moreover, the trends at the mRNA and protein levels were consistent (Figure 1B,C). Thus, the NLRP3 inflammasome was assembled, NLRP3 protein activation enhanced, and Caspase-1 cleaved into the p22 active form. The NF-κB p65 signaling pathway protein also was activated significantly during the course of infection (Figure 1D). These results indicate that the infection with *E. coli* causes inflammation in primary bEECs.

### 2.2. E. Coli Increases Cellular Ca^2+^ and Promotes Mitochondrial Dysfunction

We analyzed alterations in cellular Ca^2+^ ions and mitochondrial dysfunction during *E. coli* infection of bEECs. Using the mitochondrial Ca^2+^ indicator Rhod-2, *E. coli* caused Ca^2+^ ion efflux from mitochondria within 20 min of infection (Figure 2A). Meanwhile, Ca^2+^ imaging with the Fluo-8 probe showed a parallel increase in cytosolic levels (Figure 2B). Thus, the efflux of mitochondrial Ca^2+^ may lead to an increase in cytoplasmic Ca^2+^. Additionally, *E. coli* infection caused swelling in mitochondria, rupture of mitochondrial membranes, and disappearance of mitochondrial cristae (Figure 2C). Moreover, *E. coli* infection simultaneously caused an increase in reactive oxygen species (ROS) (Figure 2D), and a decrease in both mitochondrial membrane potential (Figure 2E) and ATP concentration (Figure 2F). These results suggest that mitochondrial dysfunction is closely related to the increase in cytosolic Ca^2+^ during *E. coli* infection of primary bEECs.

### 2.3. Cytoplasmic Ca^2+^ Regulates Mitochondrial Function

We next aimed to verify the effect of *E. coli* on cytoplasmic Ca^2+^ and explore further the regulatory role of Ca^2+^ in mitochondrial function. Using the mitochondrial Ca^2+^ sensitive fluorescent reporter Rhod-2, the results confirmed that infection of primary bEECs with *E. coli* resulted in a sharp loss of mitochondrial Ca^2+^ levels, but that this loss was not blocked by pre-treatment with the extracellular chelating agent EGTA. In contrast, pre-treatment with the permeable free Ca^2+^ chelator BAPTA-AM or IP3 receptor inhibitor 2-APB that blocks release of calcium from store-operated channels had no effect on mitochondrial Ca^2+^ levels following subsequent *E. coli* infection (Figure 3A). The efflux of mitochondrial Ca^2+^ during infection may cause a parallel increase in cytoplasmic Ca^2+^. Using the cytosolic Ca^2+^ reporter Fluo-8, we observed that, although infection with *E. coli* with or without pre-treatment with EGTA resulted in significant increases in cytosolic Ca^2+^ levels, infection with *E. coli* after pre-treatment with BAPTA-AM or 2-APB led to low levels of Ca^2+^ in the cytosol (Figure 3B). Of course, there was no doubt that the EGTA, BAPTA-AM, and 2-APT we have purchased could fully perform their function in primary bEECs (Appendix A). Furthermore, we also examined the levels of ROS (Figure 3C), membrane potential (Figure 3D), and ATP concentrations (Figure 3E) during bacterial infection after pretreatment with EGTA, the functional reactive oxygen species scavenger NAC (Appendix A), BAPTA-AM, or 2-APB. Infection with *E. coli* with or without pre-treatment with EGTA elicited no significant change in ROS levels, membrane potential, or ATP concentrations, whereas the ROS levels decreased significantly in bEECs infected with *E. coli* and pre-treated with NAC, BAPTA-AM, or 2-APB; the membrane potential and cellular ATP concentrations showed modest increases in these three cases. These results indicate that the increase in cytosolic Ca^2+^ during *E. coli* infection of bEECs is caused mainly by the release of Ca^2+^ stored in the ER and mitochondria and independent of extracellular Ca^2+^, and that cytoplasmic Ca^2+^ levels are involved in the regulation of mitochondrial function.

### 2.4. Cytoplasmic Ca^2+^ Promotes Mitochondrial Damage Caused by E. Coli

To further assess the role of cytoplasmic Ca^2+^ in regulation of mitochondrial respiratory chain transmission, mtDNA replication, and mitochondrial biogenesis, we examined the expression of relevant genes in primary bEECs following pretreatment with EGTA, NAC, BAPTA-AM, or 2-APB prior to *E. coli* infection. Cytosolic cytochrome c protein levels increased after *E. coli* infection, which indicated that the infection affected mitochondrial respiratory chain transmission (Figure 4A). However, pretreatment with NAC, BAPTA-AM, or 2-APB blocked cytochrome c release from mitochondria. In contrast, pretreatment with EGTA did not reduce the release of cytochrome c following *E. coli* infection. Moreover, expression of the *mtND1*-*mtND6* genes assessed by real-time quantitative PCR increased upon *E. coli* infection, both with and without EGTA pretreatment, which was restored by pretreatment with NAC, BAPTA-AM, or 2-APB (Figure 4B,C). As the replication of mtDNA is directed by genes located on the nuclear genome, the expression of nuclear genes that control mtDNA replication was analyzed following infection. Infection with *E. coli* increased expression of *polymerase gamma* (*POLG*), *single-stranded DNA-binding protein* (*SSBP1*), *TWINKLE* (*TWNK*), and *DNA topoisomerase 1* (*TOP1*) genes. These increases were inhibited fully or partially by EGTA, NAC, BAPTA-AM, or 2-APB pretreatment (Figure 4D). Mitochondrial biogenesis is mediated by the integration of multiple transcriptional pathways that control both nuclear and mitochondrial gene expression. Therefore, we analyzed the expression of *peroxisome proliferator-activated receptor gamma coactivator 1 alpha* (*PGC1α*), *mitochondrial transcription factor A* (*TFAM*), and *nuclear respiratory factor 1* (*NRF1*) genes in these pathways. The results were consistent with mtDNA and nuclear genes that control mitochondrial biogenesis (Figure 4E). In summary, the preceding experiments demonstrate that cytoplasmic Ca^2+^ is a key factor in regulating mitochondrial damage during *E. coli* infection.

### 2.5. The Inflammatory Response Induced by E. Coli Requires ROS Production and Release of Ca2^+^

The inflammatory response induced by cellular infection with *E. coli* was examined. Primary bEECs infected with *E. coli* demonstrated increased expression of pre-IL-1β and pre-IL-18 cytokines in both mRNA and protein levels. Pretreatment with the extracellular Ca^2+^ chelator EGTA did not ameliorate these increases. However, pretreatment with the ROS scavenger NAC, permeable free Ca^2+^ chelator BAPTA-AM, or IP3 receptor inhibitor 2-APB effectively reduced the expression of those inflammatory factors (Figure 5A–C). We next examined the signaling pathway proteins that regulate the secretion of inflammatory factors IL-1β and IL-18. The NLRP3 inflammasome was assembled and p65 signaling pathway was activated in cells infected with *E. coli* with or without EGTA pretreatment, whereas pretreatment with NAC, BAPTA-AM, or 2-APB led to low levels of NLPR3, cleaved Caspase-1 p22, and NF-κB p65 (Figure 5D). These results suggest that the production of ROS and release of cytosolic Ca^2+^ are key factors in inducing inflammatory responses in primary bEECs that are infected with *E. coli*.

### 2.6. ERS Is Involved in the Regulation of Bovine Endometritis

Numerous studies have demonstrated that ERS is involved in the regulation of inflammatory responses [20,21]. In order to analyze any correlation between bovine endometritis and ERS, we collected bovine uterine tissues with different degrees of inflammation and detected the expression of ERS marker genes. The degree of inflammation of the uterine tissue first was assessed by visual inspection (Figure 6A), HE staining of uterine tissue (Figure 6B), and Diff staining of uterine cavity mucus (Figure 6C). Subsequently, uterine tissues were divided into healthy endometrium (Figure 6a1,b1,c1), tissues with moderate endometritis (Figure 6a2,b2,c2), and tissues with severe endometritis (Figure 6a3,b3,c3). The proportions of granulocytes in moderate (Figure 6b2) and severe endometritis tissue (Figure 6b3) were significantly increased and a large number of lymphocytes were infiltrated in both cases compared with healthy endometrium tissue (Figure 6b1). Additionally, uterine tissue with severe endometritis was accompanied by endometrial gland damage (Figure 6b3), which also was evident with Diff staining of uterine cavity mucus (Figure 6c1–c3). Immunohistochemistry staining was performed to analyze the expression of the ERS marker gene GRP78 which showed that, compared with healthy endometrium, the expression of GRP78 was higher in tissues with moderate and severe endometritis, specially in endometrial cavity epithelium and glandular epithelial cells (Figure 6D). The expression levels of GRP78 and UPR signaling pathway-related proteins (p-IRE1, p-eif2α, and ATF6) in uterine tissue increased significantly with the aggravation of inflammation (Figure 6E). Overall, these results suggest that ERS is involved in the regulation of bovine endometritis.

### 2.7. ERS Involvement in Regulating the E. Coli-Induced Inflammatory Response Depends on Cytoplasmic Ca^2+^ Levels

*E. coli* is the main pathogen in bovine endometritis. In view of the results described in the preceding section, which showed that primary bEECs infected with *E. coli* for 6 h significantly increased the expression of ERS-related proteins (Figure 7A), we speculated that ERS was involved in regulating the inflammatory response induced by *E. coli* in these cells. Considering that the increase in cytoplasmic Ca^2+^ levels is due mainly to the release of Ca^2+^ stored in the ER, we speculated that cytoplasmic Ca^2+^ was involved in regulating ERS during the process of *E. coli*-induced inflammatory response. In support of this hypothesis, primary bEECs infected with *E. coli* showed increased expression of ERS-related proteins (GRP78, p-IRE1, p-eif2α, and ATF6), which was not affected by pretreatment with EGTA. In contrast, pretreatment with NAC, BAPTA-AM, or 2-APB restored protein levels to the levels without infection (Figure 7B). Therefore, we conclude that cytosolic Ca^2+^ is an important factor for regulating ERS in *E. coli-*induced inflammation in primary bEECs.

### 2.8. Enhancement of Autophagic Activity with 2-APB Reduces the E. Coli-Induced Damage Response

Activation of autophagy has been reported to attenuate the cellular inflammatory injury response [22]. LC3II is implicated in autophagy substrate selection and autophagosome biogenesis, whereas p62 is a stress-inducible selective autophagy receptor. Consistent with induced inflammation, *E. coli* inhibited autophagy activation in primary bEECs as indicated by reduced LC3II and increased p62 levels (Figure 8A). Pretreatment with 2-APB significantly increased the level of LC3II and decreased the expression of p62, thereby showing activation of autophagy during infection (Figure 8A). To further assess the role of autophagy in cytosolic Ca^2+^-mediated activation of inflammatory responses, primary bEECs were pretreated with the autophagy activator rapamycin followed by *E. coli* infection. Rapamycin and 2-APB alone or in combination displayed potent inhibitory effects on the release of inflammatory factors IL-1β and IL-18, as well as on expression of the signaling pathway proteins NF-κB p65, NLRP3, and cleaved Caspase-1 p22 (Figure 8B–E).

Additionally, activation of autophagy induced by rapamycin and/or 2-APB inhibited the release of cytochrome c, ATP, ERS-related proteins (GRP78, p-IRE1, p-eif2α, and ATF6), and cytoplasmic calcium ions Ca^2+^ (Figure 9). These data show that 2-APB enhances autophagy and attenuates the inflammatory damage responses induced by *E. coli* infection.

## 3. Discussion

Activation of innate immune signaling pathways depends on host cell recognition of pathogenic microorganisms. In the case of pathogen recognition by endometrium, the immune system releases an array of inflammatory and/or pro-inflammatory cytokines, including IL-1β, IL-6, TNF-α, and IL-18 [23,24]. Enterohemolysin produced by *E. coli* O157:H7 induces IL-1β production in human macrophages [25]. Additionally, numerous studies have shown that the NLRP3 inflammasome is involved in cellular inflammatory responses and helps to recognize pathogen infection, thereby promoting the maturation of IL-1β and IL-18 in macrophages [26,27]. In agreement with previous evidence [28], our study showed that infection of primary bEECs with *E. coli* significantly increased the expression of IL-1β and IL-18 and that the NLRP3 inflammasome simultaneously was assembled, which manifested as NLRP3 activation and cleavage of Caspase-1 into its p22 active form. These results are in agreement with the activation of Caspase-1 that is induced by *Salmonella* in intestinal epithelial cells [29]. Additionally, nuclear factor NF-κB is a crucial component of a prototypical proinflammatory signaling pathway, which is involved in the macrophage-driven inflammatory reaction and activation of the NLRP3 inflammasome [30,31,32]. Our results confirmed that NF-κB p65 was significantly activated in the inflammatory response during infection of bEECs by *E. coli*.

Calcium ions play a key role as a second messenger in the regulation of basic cellular functions, including signal transduction, gene expression, muscle contraction, hormone synthesis and secretion, and neurotransmitter release [33]. Notably, numerous studies indicated that *Listeria* and *Salmonella typhimurium* promote invasion of host cells by affecting cytosolic Ca^2+^ levels [34,35]. The Map effector protein of enteropathogenic *E. coli* triggers calcium mobilization by targeting mitochondria [36]. This observation is consistent with our results which show that increases in cytosolic Ca^2+^ induced by *E. coli* infection were associated with mitochondrial Ca^2+^ release. In addition, numerous inflammatory diseases induce mitochondrial damage and dysfunction. For example, the mitochondrial membrane potential was significantly reduced in mice with pancreatitis [14], electron transport chain of Caco-2 cells was disrupted by infection with *E. coli* [37], and large amounts of mtDNA were released in mouse models of acute nephritis [38]. Our results demonstrated that infection caused mitochondrial swelling, rupture of mitochondrial membranes, and the disappearance of mitochondrial cristae, which indicate that mitochondria were damaged. Thus, combined with the increase in ROS, decreased membrane potential, and significantly decreased ATP production, we demonstrated that inflammatory injury caused by *E. coli* leads to mitochondrial dysfunction in primary bEECs.

Intracellular calcium mobilization acts as an important signaling event during the course of pathogen infection [39]. Changes in cytosolic Ca^2+^ generally correlate with extracellular Ca^2+^ concentration and intracellular Ca^2+^ storage. Intracellular Ca^2+^ is stored mainly in the mitochondria and ER [11,39,40,41]. Human brain microvascular endothelial cells infected with *Neisseria meningitidis* release Ca^2+^ stored in the ER [39]. Additionally, *E. coli* increases cellular calcium levels by triggering extracellular calcium influx in murine bone marrow derived macrophages [40]. In this study, Ca^2+^ efflux induced by *E. coli* was blocked almost completely in the presence of BAPTA-AM or 2-APB. However, the presence of EGTA in the cell culture medium did not alter cytosolic Ca^2+^ levels. As *E. coli* infection reduced mitochondrial Ca^2+^, we propose that the increase in cytosolic Ca^2+^ during infection is caused principally by the release of calcium from ER and mitochondria irrespective of extracellular Ca^2+^ levels. As this suggestion deviates from the data presented in other studies summarized above, we speculate that the increase in cytosolic Ca2+ during infection may be related to the excessive release of intracellular stored calcium ions. ROS and antioxidant systems are in dynamic balance to maintain homeostasis under normal circumstances. However, when redox system homeostasis is perturbed, mitochondrial damage and energy metabolism disorders result, which eventually will lead to cell damage [42]. It has been reported previously that fluctuations in cytoplasmic and mitochondrial Ca^2+^ may unbalance the antioxidant system [14]. Consistent with our results, the ROS scavenger NAC and cytosolic Ca^2+^ regulators BAPTA-AM and 2-APB effectively restored mitochondrial function.

The integrity of the mitochondrial respiratory chain is a prerequisite for maintaining mitochondrial function. Cytochrome c is a crucial factor in the respiratory chain [43]. Infection with *E. coli* activated cytochrome c release in the host intestinal epithelial cells, thereby promoting mitochondrial dysfunction [37]. Moreover, a regulatory relationship was proposed between calcium ions and cytochrome c during Epstein Barr virus infection [44]. In the current study, cytochrome c levels increased after infection, which indicates that *E. coli* affected mitochondrial complex IV is the regulatory center of oxidative phosphorylation. This effect was restored by cytosolic Ca^2+^ regulators. These results further demonstrate that the release of cytochrome c is affected by the level of cytosolic calcium ions during pathogen infection. Mitochondria, as the power station of cells, produce ATP through oxidative phosphorylation to maintain normal cellular activities [43,45]. This process depends on the expression of genes on the mtDNA genome. Damaged mitochondria release mtDNA into the cytosol, which activates the NLRP3 inflammasome [46]. Our results showed that infection with *E. coli* increased the levels of proteins encoded by mtND1-mtND6 proteins and that this increase was reduced by NAC, BAPTA-AM, or 2-APB. Immune cells that are deficient in proteins required for mtDNA replication lose mtDNA, which results in defective inflammatory activation [47]. Our findings are in agreement with these observations: the reduction in mtDNA caused by NAC, BAPTA-AM, or 2-APB during *E. coli* infection also reduced the expression of inflammatory factors IL-1β and IL-18. In parallel, the levels of NLRP3 inflammasome and NF-κB p65 also decreased. The replication of *mtDNA* depends on a complete set of nuclear genes that encodes its production [48]. Thus, nuclear genes that control the replication of *mtDNA* explain the occurrence, development, and frequency of mitochondrial diseases [49]. Here, the expression of nuclear genes *POLG*, *SSBP1*, *TWNK*, and *TOP1* during *E. coli* infection was consistent with that of mtDNA.

Infection with *E. coli* alters mitochondrial biogenesis as well as *mtDNA* replication. *PGC1α* is the master regulator of mitochondrial biogenesis and considered a promising target for the treatment of mitochondrial diseases. Cardiac *PGC1α* overexpression leads to severe structural abnormalities in cardiomyocytes, and neuronal overexpression similarly leads to impaired neurological function [50,51]. Accordingly, we found that infection with *E. coli* induced *PGC1α* activation, which may be detrimental to normal cellular activity in primary bEECs. Nuclear-encoded electron transport chain components are activated by *NRF1* [45] and chronic inflammation increases NRF1 expression in IL-6-treated adipocytes, which leads to mitochondrial dysfunction [52]. Our results showed that infection also enhances *NRF1* expression, which may have negative effects on host cells. *TFAM* plays a central role in the *mtDNA* stress-mediated inflammatory response and is involved in a variety of neurodegenerative conditions, including Alzheimer’s disease, Huntington’s disease, and Parkinson’s disease [53]. Our results demonstrated that *TFAM* is involved in *E. coli*-induced mitochondrial oxidative damage. Consistent with previous results, decreased cytosolic Ca^2+^ levels ameliorated mitochondrial biogenesis. The above results collectively indicate that cytoplasmic Ca^2+^ is a key factor in regulating mitochondrial damage and inflammation during *E. coli* infection.

Bovine endometritis is a major economic problem globally and manifests histologically as epithelial disintegration, lymphocyte aggregation, and inflammatory cell infiltration [54]. Numerous studies have used endometrial histopathology to determine the degree of uterine inflammation [55,56]. In this study, HE staining of uterine tissue and Diff staining of uterine cavity mucus were used to divide uterine tissue into samples with healthy endometrium, moderate endometritis, and severe endometritis depending on infiltration of granulocytes and lymphocytes. ERS is involved in the endometrial inflammatory response in goat and mouse uterine tissue [20,21]. Our study also demonstrated that the ERS marker protein GRP78, and p-IRE1, p-eif2α and ATF6 implicated in UPR are involved in the regulation of uterine tissue inflammation in bovine. As the main pathogen that mediates bovine endometritis [4], *E. coli* incontrovertibly causes an inflammatory response in primary bEECs. Similar to the results in bovine uterine tissue, ERS was involved in regulating the endometrial inflammatory injury response induced by *E. coli*, although there are differences in the regulatory mechanisms of ATF6 in the two tissues. The integration and coordination of Ca^2+^ transport molecules, Ca^2+^ buffers, and Ca^2+^ sensors maintain cellular Ca^2+^ homeostasis. These molecules are closely associated with different cellular components, most notably the ER and mitochondria [11]. Other studies have shown that LPS modulates the levels of cytoplasmic calcium, as well as ERS in human uterine smooth muscle cells. Additionally, prechelation with BAPTA-AM effectively decreased the expression of p-IRE1α and XBP1s in the myocytes. However, the levels of GRP78 protein did not change [57]. These observations mirror the data presented here, although slight difference in the regulatory pathways exist. Our results showed that 2-APB and BATPA-AM reduced the expression of GRP78 and UPR-related proteins (p-IRE1, p-eif2α and ATF6) during *E. coli* infection, which implies that cytosolic Ca^2+^ may regulate the degree of inflammation by regulating ERS. Additionally, oxidative stress and inflammation interact via ERS, and ROS overproduction activates the UPR, which disrupts ER homeostasis and promotes inflammation and cell death [58], which explains our observation that NAC reduces the expression of ERS-related proteins during *E. coli* infection. Therefore, we speculate that cytoplasmic Ca^2+^ directly or indirectly through mitochondrial oxidative damage regulates ERS during *E. coli*-induced inflammatory injury of primary bEECs, thereby regulating the inflammatory response.

Cellular autophagy regulates innate immune responses by rearranging organelles and proteins [22]. Similarly, damaged mitochondria need to be removed by autophagy. However, inhibition of autophagy results in the abnormal accumulation of damaged mitochondria, which leads to inflammasome activation and inflammatory responses [59]. Accordingly, we found *E. coli* inhibited autophagy in primary bEECs, which may relate to enhanced inflammasome activity. Several studies reported that Ca^2+^ may inhibit autophagy through the inositol 1,4,5-trisphosphate receptor (IP3R), the beclin1-Bcl-2 complex, and the AMPK-mTOR pathway [60]. Similarly, our study showed that 2-APB may reduce mitochondrial damage and ERS by limiting cytosolic Ca^2+^ levels, thereby reducing the release of inflammatory factors which, in turn, increases autophagy levels. Thus, impaired autophagy is a key pathogenic event in *E. coli*-induced inflammatory injury of primary bEECs.

In conclusion, we demonstrated here that mitochondrial dysfunction and ERS in endometritis result from cytosolic Ca^2+^ overload, and that impaired autophagy is a major downstream event of mitochondrial damage and ERS. Restoring autophagic functions may serve as a strategy for the treatment of endometritis.

## 4. Materials and Methods

### 4.1. Bovine Uterine Tissue Collection

Infertile Holstein cattle, from which underlying diseases such as mastitis were excluded, were selected. Fresh bovine uteri were collected from slaughterhouses, placed in pre-chilled saline, and immediately shipped back to the laboratory. A portion of uterine mucus was scraped for subsequent Diff staining. Additionally, 4 × 4 mm tissue samples were collected. One sample was placed in 4% paraformaldehyde for HE staining and immunohistochemical testing, and a second sample was used for total protein extraction.

### 4.2. Bacterial Strains, Cell Culture and Growth Conditions

The *E. coli* used in this study were isolated from bovine uteri with endometritis and cultured in Luria-Bertani (LB) agar plates or LB broth. The identification of *E. coli* was based on colony characteristics and 16S rRNA sequence alignment. Primary bEECs were isolated and cultured from healthy bovine uteri, which were in the early-to-middle stages of the estrus cycle. Primary bEECs were propagated in DMEM/F12 medium (Hyclone, Logan, UT, USA) supplemented with 10% FBS (ZETA, Lower Gwynedd Township, PA, USA) in a humidified atmosphere with 5% CO_2_ at 37 °C. All the experimental cells were in the fifth passage. *E. coli* was inoculated at a multiple of infection of 10 to 1 for 6 h when the growth density of bEECs was >90%.

### 4.3. Real-Time Quantitative PCR

Total cellular RNA was extracted by RNAiso Plus (Takara, Gunma, Japan), and then reverse transcribed into cDNA using Evo M-MLV RT Kit (AG Bio, Changsha, China). The cDNA was used as a template for real-time quantitative PCR with the SYBR Green Pro Taq HS Mix kit (AG Bio, Changsha, China) in a Bio-Rad CFX96 instrument (Bio-Rad, Hercules, CA, USA) to measure mRNA levels using the manufacturer’s protocol. Primer sequences are shown in Appendix A. The 2^−ΔΔCt^ method was used to analyze expression levels. All samples were normalized by using the *GAPDH* gene as a control.

### 4.4. Western Blot Analysis

Western blot analysis was conducted according to previously described procedures [61] with the following antibodies: IL-1β (Proteintech, Beijing, China); IL-18 (Abcam, Cambridge, UK); NLPR3 (Abcam, Cambridge, UK); Caspase-1 (Abcam, Cambridge, UK); NF-κB p65 (Abcam, Cambridge, UK); cytochrome C (Abcam, Cambridge, UK); GRP78 (Proteintech, Beijing, China); p-IRE1 (Abcam, Cambridge, UK); p-eif2α (Abways, Shanghai, China); ATF6 (Abcam, Cambridge, UK); P62 (Abcam, Cambridge, UK); LC3 (Abcam, Cambridge, UK); and, β-actin (Proteintech, Beijing, China). Binding of antibodies was detected using HRP-labeled goat anti-rabbit (ZHHC, Shaanxi, China) or goat anti-mouse (ZHHC, Shaanxi, China) immuno-globulin. The bands were imaged subsequently using a chemiluminescent gel imaging system (Bio-Rad, Hercules, CA, USA).

### 4.5. Real-Time Cellular Imaging

Real-time cellular imaging was performed with either mitochondrial Ca^2+^ indicator Rhod-2 AM (Abcam, Cambridge, UK) or the cytoplasmic Ca^2+^ indicator Fluo-8 AM (Abcam, Cambridge, UK), according to the manufacturer’s instructions. Briefly, bEECs were seeded into 6-well plates and cultured for two days when the cell growth reached 70%. Cells were washed with phosphate buffer solution (PBS) for three times and then incubated with either Rhod-2 AM (1 μM) or Fluo-8 AM (1 μM) for 20 min at 37 °C in a CO_2_ incubator. After washing three times with PBS, cells were infected with *E. coli* and subjected to time-lapse confocal imaging using a spinning disk confocal microscope (Andorra, UK).

### 4.6. Transmission Electron Microscopy

Cells were seeded in 90 mm dishes until the concentration was >90% and then infected with *E. coli* and treated with pharmaceutical inhibitors. The culture medium was discarded and an electron microscope fixative solution was added at 4 °C for 2–4 h. The cells were harvested at low centrifugation speed, coated with 1% agarose, and washed with 0.1 M PBS (pH 7.4) three times for 15 min. 1% osmic acid·0.1 M PBS (pH 7.4) was added to the treated cells at 20 °C for 2 h and cells were washed three times with 0.1 M PBS (pH 7.4) for 15 min. Upward dehydration was performed sequentially with 50, 70, 80, 90, 95, and 100% alcohol, followed by two treatments with 100% acetone for 15 min. The samples were infiltrated with acetone and 812 embedding medium at a ratio of 1:1 for 4 h, followed by incubation at a ratio of 2:1 with pure 812 embedding medium for 8 h. Pure 812 embedding medium was poured into an embedding plate and the sample was inserted into the embedding plate at 37 °C overnight followed by incubation at 60 °C for polymerization for 48 h. Then, 60–80 nm ultrathin sections were sliced with an ultramicrotome, double-stained with uranium and lead, and observed under an electron microscope (FEI, Hillsboro, OR, USA). Images were collected for subsequent analysis.

### 4.7. Cellular ROS Detection

Cellular ROS detection was performed with the Reactive Oxygen Species Assay Kit (Beyotime, Shanghai, China), according to the manufacturer’s instructions. Briefly, cells were seeded in confocal dishes until growth was >70%. DCFH-DA was diluted 1:1000 in serum-free medium to a final concentration of 10 μmol/L. The cell culture medium was removed and an appropriate volume of diluted DCFH-DA was added that was sufficient to cover the cells. At least 1 mL of diluted DCFH-DA was typically added to each well in a six-well plate. Cells were incubated at 37 °C for 20 min, and then washed three times with serum-free cell culture medium to remove excess DCFH-DA. Cells were observed with a laser confocal microscope (Nikon, Tokyo, Japan).

### 4.8. Mitochondrial Membrane Potential and Apoptosis Detection

Detection of mitochondrial perturbation was performed with the Mitochondrial Membrane Potential and Apoptosis Detection Kit with Mito-Tracker Red CMXRos and Annexin V-FITC (Beyotime, Shanghai, China), according to the instructions. Briefly, cells were seeded in confocal dishes until growth was >70%. The cell culture medium was removed by aspiration and cells were washed once with PBS. Annexin V-FITC (188 µL), Annexin V-FITC (5 µL), and Mito-Tracker Red CMXRos staining solution (2 µL) were added in sequence and mixed gently. Samples were incubated in the dark for 20–30 min at 20–25 °C and then were observed with a laser confocal microscope (Nikon, Tokyo, Japan).

### 4.9. Cellular ATP Detection

Cellular ATP was detected with the ATP Assay Kit (Beyotime, Shanghai, China) according to the manufacturer’s instructions. Briefly, cells were grown in 6-well plates until the concentration reached 90%. The culture medium was removed by suction and 200 μL of lysis buffer was added with repeated pipetting to fully lyse the cells. Cells were centrifuged at 12,000× *g* for 5 min at 4 °C, and the supernatant was used for subsequent analysis. One hundred microliters of working ATP detection solution were added to the sample followed by incubation at room temperature for 3–5 min to consume the background ATP. An additional 20 μL of solution was added to reduce the background. The RLU values were measured with a multifunctional enzyme label instrument (BioTek, Winooski, VE, USA) and the concentration of ATP in the sample was calculated using a standard curve of 0.01, 0.03, 0.1, 0.3, 1, 3, and 10 μM ATP. The value was converted to nmol/mg protein according to the protein concentration.

### 4.10. Diff Staining

Bovine uterine mucus was stained with a Diff-Quick Stain Kit (Solarbio, Beijing, China). Briefly, the mucus was applied to a glass slide and treated with staining solution for 5–10 s. The slide was agitated up and down to ensure that the staining solution was evenly distributed. This step was repeated with staining solution for 10–20 s; the sample was washed with water, and then observed immediately by microscopy (Nikon, Tokyo, Japan).

### 4.11. Hematoxylin-Eosin (HE) Stain

Bovine uterine tissue was stained with the HE Stain Kit (Solarbio, Beijing, China). Fresh uterine tissue was fixed in 4% paraformaldehyde, embedded in conventional paraffin, and sliced at 3–8 μm. The sample was deparaffinized in xylene twice for 5–10 min, treated serially with ethanol (100, 95, 85, and 75%) with rehydration for 3 min per gradient, and then soaked in distilled water for 2 min. The sample was stained with hematoxylin staining solution for 10 min and washed with distilled water to remove floating color. The sample was stained with eosin for 1 min, dehydrated quickly by immersion in 75, 85, 95, and 100% ethanol for 2–3 s, followed by 100% ethanol for 1 min, and transparent xylene twice for 1 min. The samples were mounted with neutral gum and observed by microscopy (Nikon, Tokyo, Japan).

### 4.12. Immunohistochemistry

Fresh uterine tissue was fixed in 4% paraformaldehyde, embedded in conventional paraffin, and sliced at 3–8 μm. After dehydration, the sections were placed in citrate buffer and heated to extract antigens. Inactivated endogenous peroxidase was added to the sections in hydrogen peroxide/methanol for 1 h followed by washing in PBS. The sections were incubated in 10% pre-immune serum (Maixin-Biotech, Fuzhou, China) for 30 min at 37 °C prior to incubation with anti-GRP78 antibody (Proteintech) for 12 h at 4 °C. Preimmune serum was used as a negative control. After washing, the sections were incubated with biotinylated antibiotic-rabbit IgG antibody (Maixin-Biotech, Fuzhou, China) for 30 min at 37 °C. Streptavidin labeled with horseradish peroxidase was added with incubation at 37 °C for 30 min. Microscopy (Nikon, Tokyo, Japan) was used to evaluate the sections after staining with DAB chromogenic solution (Maixin-Biotech, Fuzhou, China).

### 4.13. Statistical Analysis

Results are reported as the arithmetic means ± S.E.M. of three samples. One-way ANOVA followed by Tukey’s post hoc test and Fisher’s LSD were used for multiple comparisons. Statistical significance was defined when the *p*-value was <0.05.

## 5. Conclusions

In this study, we demonstrated that autophagy is a key regulatory pathway in the inflammatory response caused by *E. coli* in primary bEECs. Restoring autophagic functions can effectively attenuate the inflammatory damage response. These observations are of particular significance for reducing the duration of bovine postpartum endometritis and promoting rapid postpartum pregnancy.

## Figures and Tables

**Figure 1 ijms-23-14174-f001:**
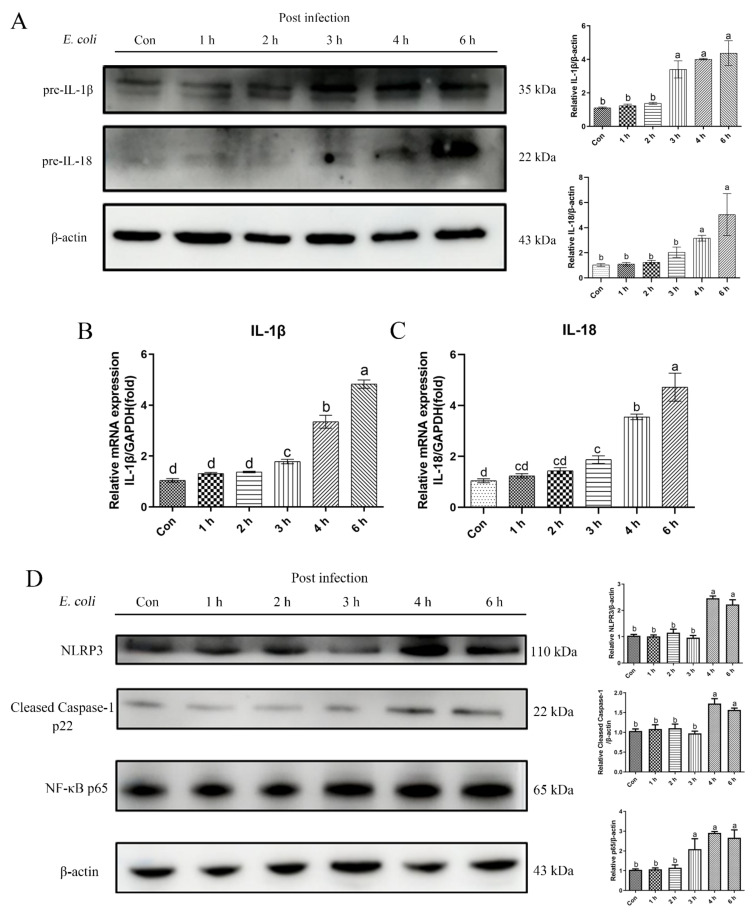
*E. coli*-induced inflammatory response in bEECs. Primary bEECs were infected with *E. coli* for 1 to 6 h, and cells were collected for protein and mRNA analyses. (**A**) Western blotting was used to analyze the pre-IL-1β and pre-IL-18 levels in whole cell lysates. (**B**) Real-time quantitative PCR analysis of the expression of *IL-1β*. (**C**) Quantification of *IL-18* expression, with *GAPDH* as a control. (**D**) The levels of NLPR3, cleaved Caspase-1, and NF-κB p65 proteins were analyzed by immunoblotting. The data are means ± SEM of three independent experiments. Bars with different letters indicate significant differences (*p* < 0.05).

**Figure 2 ijms-23-14174-f002:**
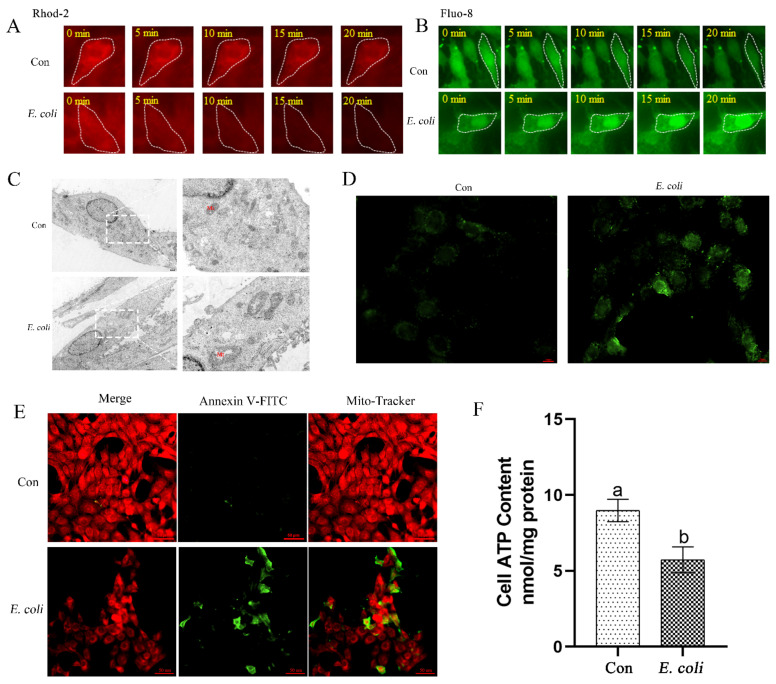
Effects of *E. coli* on calcium mobilization and mitochondrial function in bEECs. Primary bEECs were either uninfected or infected with *E. coli* and cytoplasmic Ca^2+^ levels and cellular mitochondria function was measured. (**A**) Mitochondrial Ca^2+^ reporter Rhod-2 was used for real-time cellular imaging to measure mitochondrial Ca^2+^ levels. The cells were infected with or without *E. coli* for up to 20 min. (**B**) Fluo-8 AM was used to monitor cytosolic Ca^2+^ levels in real time. Cells were exposed or not exposed to *E. coli* for up to 20 min. (**C**) The mitochondria in bEECs were analyzed by transmission electron microscopy after infection with *E. coli* for 6 h. (**D**) ROS was measured after infection with *E. coli* for 6 h. Scale bars = 10 μm. (**E**) Cells were infected with *E. coli* for 6 h, and membrane potential was detected by Mito-Tracker Red CMXRos. Scale bars = 50 μm. (**F**) The ATP concentration was detected using an ATP assay kit after infection for 6 h. Con, no infection control; Mt, mitochondria. The data are means ± SEM of three independent experiments. Bars with different letters indicate significant differences (*p* < 0.05).

**Figure 3 ijms-23-14174-f003:**
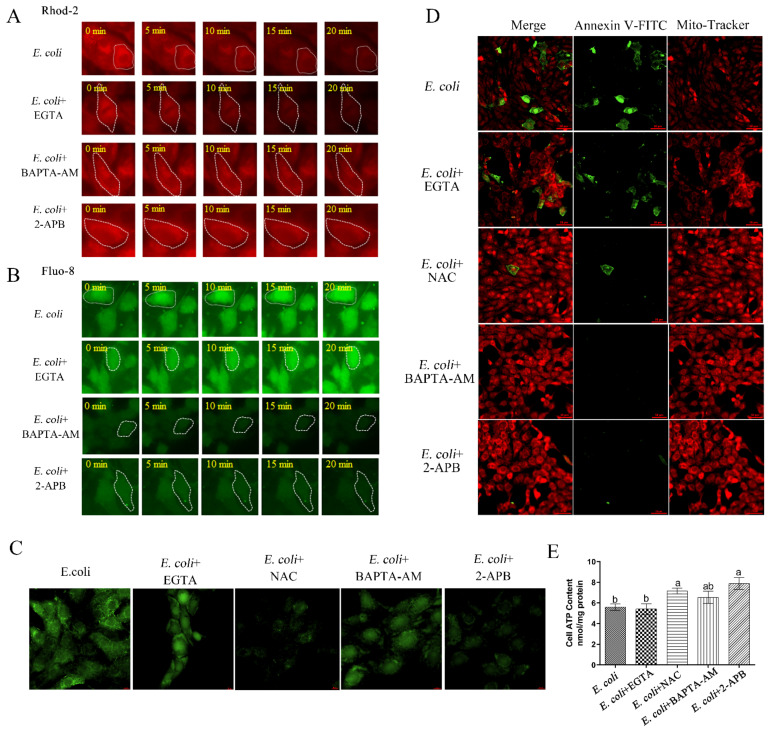
The role of cytoplasmic Ca^2+^ in mitochondrial function. Primary bEECs were pre-treated with EGTA (5 mM), BAPTA-AM (10 μM), 2-APB (50 μM), or NAC (8 mM) before *E. coli* infection. EGTA is a Ca^2+^ chelator that captures extracellular Ca^2+^. BAPTA-AM is a selective, permeable free Ca^2+^ chelator that sequesters cytoplasmic Ca^2+^. 2-APB inhibits IP3 receptors and blocks release of calcium from store-operated calcium channels. The anti-oxidant properties of NAC are thought to combat some of the effects of oxidative stress. (**A**) Live-cell imaging was used to monitor mitochondrial Ca^2+^ levels. (**B**) Fluo-8 AM was used to monitor cytosolic Ca^2+^ levels in real time. Cells were infected with *E. coli* for up to 20 min. (**C**) ROS levels were detected by DCFH-DA fluorescence after 6 h of infection. Scale bars = 10 μm. (**D**) Cells were exposed to *E. coli* for 6 h and Mito-Tracker Red CMXRos was used to monitor membrane potential. Scale bars = 50 μm. (**E**) Cellular ATP content was detected by an ATP assay kit after infection for 6 h. The data are means ± SEM of three independent experiments. Bars with different letters indicate significant differences (*p* < 0.05).

**Figure 4 ijms-23-14174-f004:**
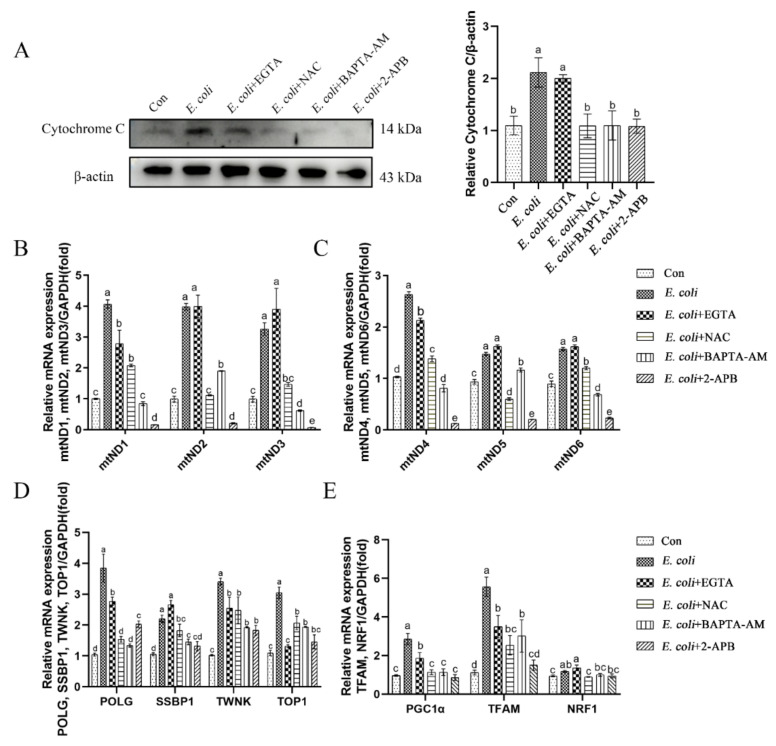
Cytoplasmic Ca^2+^ regulates mitochondrial damage during *E. coli* infection. Primary bEECs were pre-treated with EGTA (5 mM), NAC (8 mM), BAPTA-AM (10 μM), or 2-APB (50 μM) before *E. coli* infection for 6 h. (**A**) Western blot analysis of the expression levels of cytochrome c. (**B**) Real-time quantitative PCR of expression of mitochondrial genome-related genes *mtND1*, *mtND2* and *mtND3*. (**C**) Real-time quantitative PCR analysis of *mtND4*, *mtND5*, and *mtND6* levels. (**D**) Nuclear genes (*POLG*, *SSBP1*, *TWNK*, and *TOP1*) that control the replication of mtDNA were detected by real-time quantitative PCR. (**E**) Relative mRNA expression of mitochondrial biogenesis-related genes (*PGC1α*, *TFAM*, and *NRF1*). The data are means ± SEM of three independent experiments. Bars with different letters indicate significant differences (*p* < 0.05).

**Figure 5 ijms-23-14174-f005:**
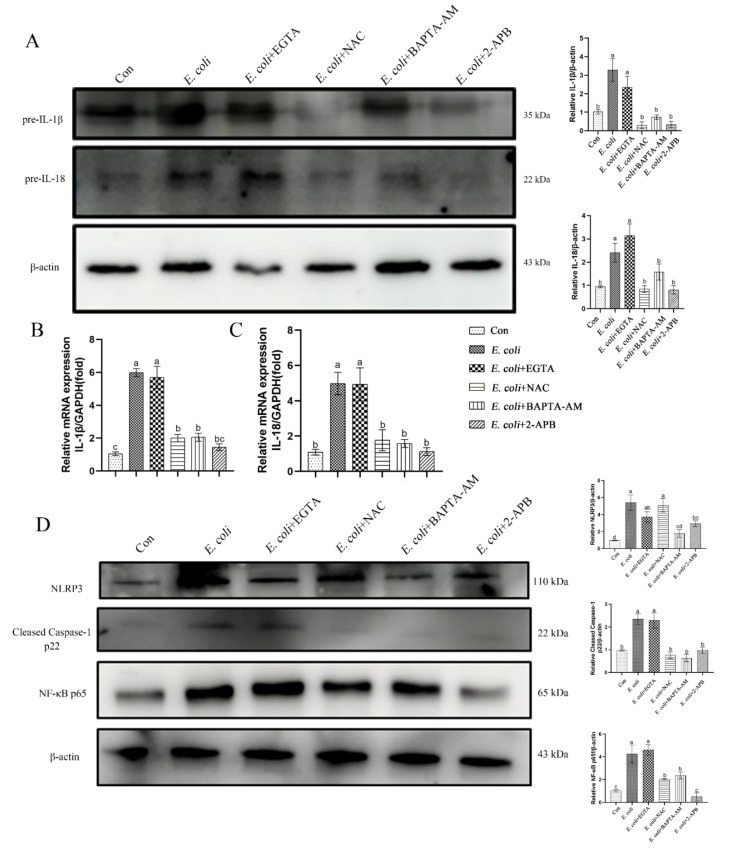
The inflammatory response induced by *E. coli* infection requires ROS production and release of Ca^2+^. Primary bEECs were pre-treated with EGTA (5 mM), NAC (8 mM), BAPTA-AM (10 μM), or 2-APB (50 μM) before *E. coli* infection for 6 h. (**A**) Immunoblotting analysis of IL-1β and pre-IL-18 cytokine proteins. (**B**) Real-time quantitative PCR of the expression of *IL-1β*. (**C**) *IL-18* was detected by real-time quantitative PCR. (**D**) Protein levels of NLPR3, cleaved Caspase-1, and NF-κB p65 were analyzed by immunoblotting. Con, no infection control. The data are means ± SEM of three independent experiments. Bars with different letters indicate significant differences (*p* < 0.05).

**Figure 6 ijms-23-14174-f006:**
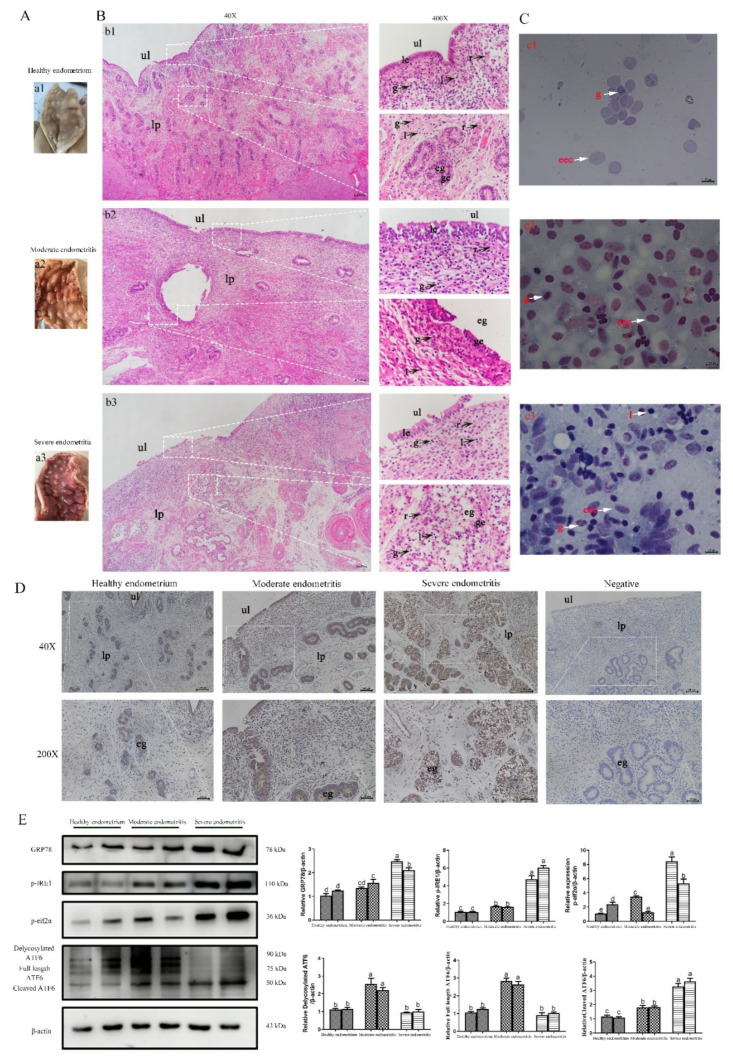
Effect of ERS in dairy bovine endometritis. (**A**) Postpartum bovine uterus was collected for visual inspection. (**B**) Hematoxylin and eosin staining was used to analyze the degree of inflammation in bovine endometrium. (**C**) Diff staining of uterine cavity mucus was used to assess the degree of inflammation of the uterine tissue. Micrographs of the bovine endometrium show healthy endometrium (**a1**,**b1**,**c1**), moderate endometritis (**a2**,**b2**,**c2**) and severe endometritis (**a3**,**b3**,**c3**). Scale bars = 200 μm. (**D**) Immunohistochemistry showing the expression of the ERS marker protein GRP78 in bovine uteri with different levels of endometritis. (**E**) The expression of ERS-related proteins (GRP78, p-IRE1, p-eif2α, and ATF6) was measured by Western blotting in bovine uteri with different levels of endometritis. ul = uterine lumen; eg = endometrial gland; le = luminal epithelium; ge = glandular epithelium; lp = lamina propria; g = granulocytes; l = lymphocytes; r = red blood cells; eec = endometrial epithelium cells. The data are means ± SEM of three independent experiments. Bars with different letters indicate significant differences (*p* < 0.05).

**Figure 7 ijms-23-14174-f007:**
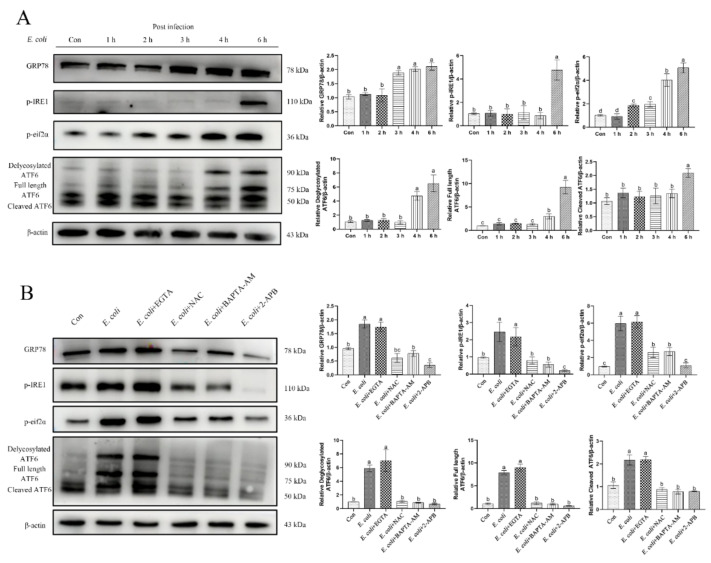
ERS is involved in regulating the *E. coli*-induced inflammatory response in bEECs. Primary bEECs were either uninfected or infected with *E. coli* for 6 h after pretreatment with EGTA (5 mM), NAC (8 mM), BAPTA-AM (10 μM), or 2-APB (50 μM). (**A**) Western blot analysis of ERS-related proteins (GRP78, p-IRE1, p-eif2α, and ATF6) during *E. coli* infection for 1–6 h. (**B**) Expression of ERS-related proteins (GRP78, p-IRE1, p-eif2α, and ATF6) measured by Western blot after 6 h of infection. The data are means ± SEM of three independent experiments. Bars with different letters indicate significant differences (*p* < 0.05).

**Figure 8 ijms-23-14174-f008:**
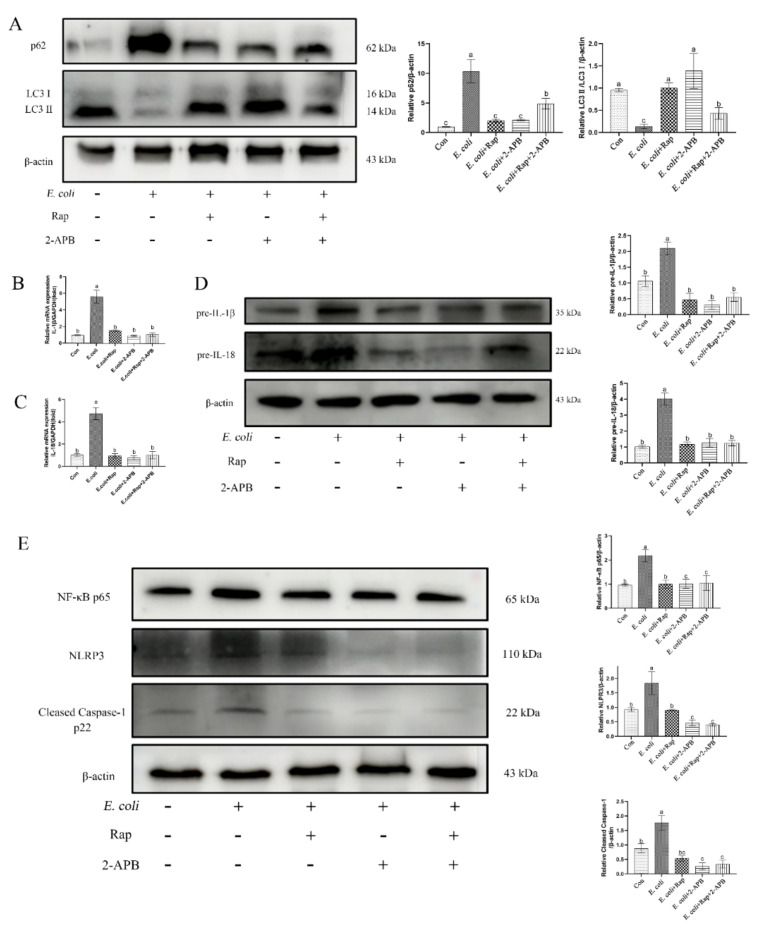
Cytoplasmic Ca^2+^ inhibits the activation of autophagy in bEECs infected with *E. coli*. Primary bEECs were either uninfected or infected with *E. coli* for 6 h after pretreatment with rapamycin (10 μM) or 2-APB (50 μM). Rapamycin is a potent and specific mTOR inhibitor that activates autophagy. (**A**) Western blot analysis of the expression of autophagy-related proteins (LC3 and p62). (**B**) Real-time quantitative PCR analysis of the expression of *IL-1β*. (**C**) *IL-18* was detected by real-time quantitative PCR. (**D**) The levels of pre-IL-1β andpre-IL-18 were analyzed by immunoblotting. (**E**) Western blotting was used to analyze the levels of NLPR3, cleaved Caspase-1, and NF-κB p65 proteins. The data are means ± SEM of three independent experiments. Bars with different letters indicate significant differences (*p* < 0.05).

**Figure 9 ijms-23-14174-f009:**
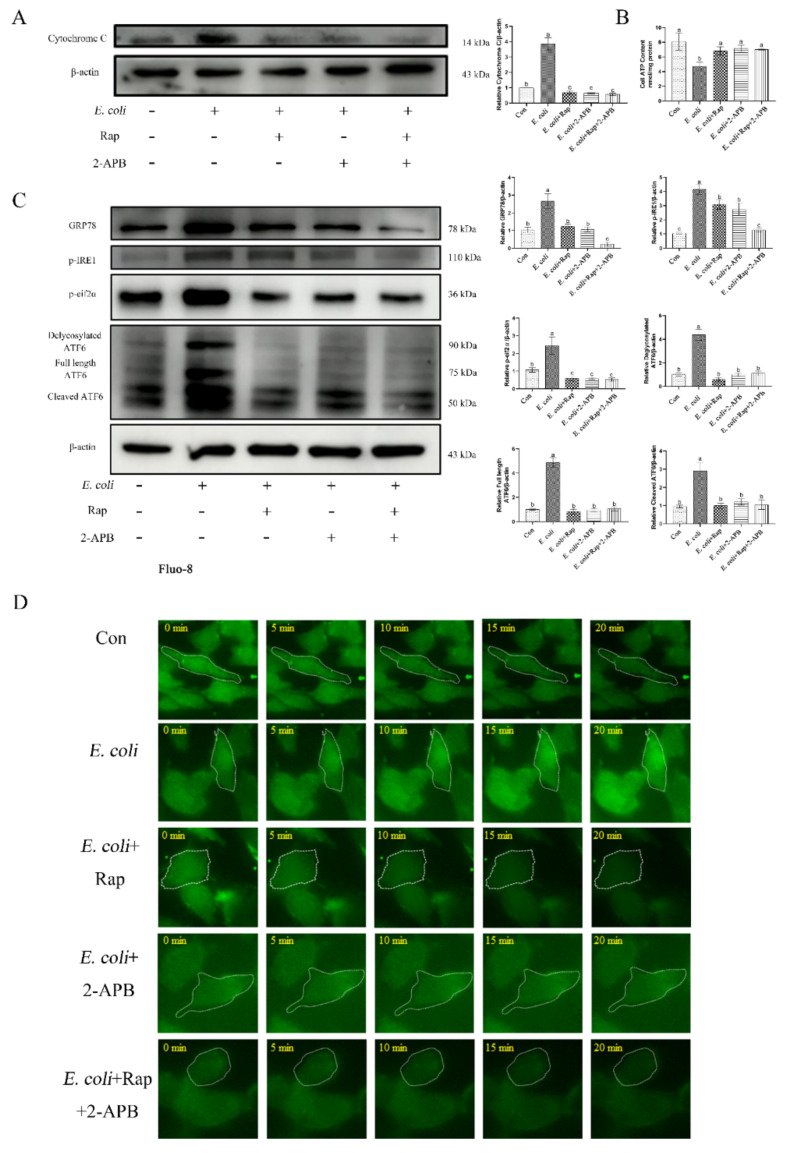
Enhancement of autophagic activity with 2-APB and rapamycin improves *E. coli*-induced damage response. Primary bEECs were either uninfected or infected with *E. coli* for 6 h after pretreatment with rapamycin (10 μM) or 2-APB (50 μM). (**A**) Western blot analysis of cytochrome c levels. (**B**) ATP content was detected by an ATP assay kit. (**C**) The expression of ERS-related proteins (GRP78, p-IRE1, p-eif2α, and ATF6) was measured by Western blot. (**D**) Fluo-8 AM was used to monitor cytosolic Ca^2+^ levels in real time. Cells were exposed to *E. coli* for up to 20 min. Con, no infection control. The data are means ± SEM of three independent experiments. Bars with different letters indicate significant differences (*p* < 0.05).

## Data Availability

Not applicable.

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
