# Peer review of "Autophagy Mediates Escherichia Coli-Induced Cellular Inflammatory Injury by Regulating Calcium Mobilization, Mitochondrial Dysfunction, and Endoplasmic Reticulum Stress"

_ijms, 2022, doi:10.3390/ijms232214174_

Round 1

Reviewer 1 Report

This is a good concise paper.

However it is only speculation, it is not valued to read.

Reviewer 2 Report

The article and the topic are very interesting. The manuscript is generally well-written and well-structured. Nevertheless, the authors should improve the discussions section by comparing their results with other results reported in the literature. In this form, the authors used mostly general information in the discussions section that should be removed since they do not represent the aim of the discussions section.

Moreover, the conclusions should focus more on highlighting the clinical relevance of this study.

Author Response

Dear Reviewer:

Thank you very much for the helpful comments on our manuscript (ID: ijms-1971059). We have read the comments carefully and have adjusted the manuscript accordingly. Our response to your specific comments are outlined below.

A revised manuscript has been uploaded to the journal website. Please feel free contact us with any further questions.

Best Regards,

---------------------------------------

Comments and Suggestions for Authors: The article and the topic are very interesting. The manuscript is generally well-written and well-structured.

Point 1: Nevertheless, the authors should improve the discussions section by comparing their results with other results reported in the literature. In this form, the authors used mostly general information in the discussions section that should be removed since they do not represent the aim of the discussions section.

Response 1: Thank you very much for your suggestion. We have improved the discussion section of the manuscript by comparing our results more fully with the reports in the literature and have removed more general information in the discussion section.

Point 2: The conclusions should focus more on highlighting the clinical relevance of this study.

Response 2: Thank you very much for your suggestion. We have rewritten the conclusions in the manuscript as follows: In this study, we demonstrated that autophagy is a key regulatory pathway in the inflammatory response caused by E. coli in primary bEECs. Restoring autophagic functions can effectively attenuate the inflammatory damage response. These observations are of particular significance for reducing the duration of bovine postpartum endometritis and for promoting rapid postpartum pregnancy.

Reviewer 3 Report

This study is being validated using cultured cells as a model system for cellular response to bacterial infection in endometritis. This reviewer thinks that this article is worth for publication after several modifications. This reviewer would like the authors to consider the following points;

(1)  How does the cellular response change in response to Multiples of infections (MOI)? Is a 10-fold increase in the number of E. coli relative to the number of cells (MOI 10) tested in this paper considerably more excessive conditions than is actually the case. How about the cellular response attenuated at MOI 0.1 and 1? Is there a response threshold?

(2)  EGTA, NAC, BAPTA-AM, and 2-APB treatment experiments should be checked for response to bEECs not infected with E. coli as control experiments.

(3)  Please add explanations for the abcd annotations in the graphs to the figure legends.

Author Response

Dear Reviewer:

Thank you very much for the helpful comments on our manuscript (ID: ijms-1971059). We have read the comments carefully and have adjusted the manuscript accordingly. Our response to your specific comments are outlined below.

A revised manuscript has been uploaded to the journal website. Please feel free contact us with any further questions.

Best Regards,

------------------------------------

Comments and Suggestions for Authors: This study is being validated using cultured cells as a model system for cellular response to bacterial infection in endometritis. This reviewer thinks that this article is worth for publication after several modifications. This reviewer would like the authors to consider the following points;

Point 1: How does the cellular response change in response to Multiples of infections (MOI)? Is a 10-fold increase in the number of E. coli relative to the number of cells (MOI 10) tested in this paper considerably more excessive conditions than is actually the case. How about the cellular response attenuated at MOI 0.1 and 1? Is there a response threshold?

Response 1: We are sorry that this problem has confused you. In fact, we screened the multiplicity of infection (MOI) of primary bovine endometrial epithelial cells (bEECs) infected with E. coli at an early stage of the study. We consulted a wide range of the literature as there are numerous differences in MOI for different cell lines, such as bone marrow-derived macrophages (BMDM) and peritoneal macrophages (MOI, 0.2, 0.5) [1], intestinal porcine epithelial cell line 1 (IPEC-1) cells (MOI, 0.1) [2], sheep endometrial epithelium cells (MOI, 1) [3], bovine mammary epithelial cells (MOI, 5) [4], and Caco-2 cells (MOI, 10) [5-7]. Some papers even report E. coli MOI of 50 or 100 for cell infection [8]. The MOI gradients we selected at the initial stage were 1:10, 1:5, 1:1, 5:1, 10:1, 20:1, 50:1 and 100:1. However, changes in cellular morphology gradually appeared as the MOI increased, and significant changes in morphology were observed when the MOI of E. coli was 20:1, 50:1, and 100:1 at which point vacuolization, shrinkage, transparency, detachment, and even dead cells appeared. The expression of NF-κB, IL-1β, and IL-18 increased with a dose-dependent effect when MOI was below 10:1, and decreased gradually when MOI exceeded 10:1. These results indicate that the MOI of E. coli at 10:1 was appropriate to establish the inflammation model of primary bEECs. Since the cells we used were primary bEECs, which may be less prone to infection than cell lines such as RAW264.7, a higher MOI was possible. Concerning a response threshold, the primary bEECs infected with E. coli indeed have a response threshold. In our study, E. coli infection significantly activated the inflammatory response at a MOI of 1 for 3 h.

Point 2: EGTA, NAC, BAPTA-AM, and 2-APB treatment experiments should be checked for response to bEECs not infected with E. coli as control experiments.

Response 2: Thank you very much for your suggestion and we are sorry that this problem has confused you. The inhibitors we used in the study were commercially available and numerous previous studies have demonstrated their function [9-12]. As the aim of the present study was to explore whether the inhibitors exerted effects during E. coli infection, we did not include groups only treated with the inhibitors.

Point 3: Please add explanations for the abcd annotations in the graphs to the figure legends.

Response 3: Thank you very much for your suggestion. We have added explanations for the abcd annotations in the figure legends in the manuscript.

  1. Kumar, P.; Saini, K.; Saini, V.; Mitchell, T. Oxalate Alters Cellular Bioenergetics, Redox Homeostasis, Antibacterial Response, and Immune Response in Macrophages. Frontiers in immunology 2021, 12, 694865, doi:10.3389/fimmu.2021.694865.
  2. Luo, X.; Wu, S.; Jia, H.; Si, X.; Song, Z.; Zhai, Z.; Bai, J.; Li, J.; Yang, Y.; Wu, Z. Resveratrol alleviates enterotoxigenic Escherichia coli K88-induced damage by regulating SIRT-1 signaling in intestinal porcine epithelial cells. Food & function 2022, 13, 7346-7360, doi:10.1039/d1fo03854k.
  3. Hu, X.; Wang, M.; Pan, Y.; Xie, Y.; Han, J.; Zhang, X.; Niayale, R.; He, H.; Li, Q.; Zhao, T.; et al. Anti-inflammatory Effect of Astragalin and Chlorogenic Acid on Escherichia coli-Induced Inflammation of Sheep Endometrial Epithelium Cells. Front Vet Sci 2020, 7, 201, doi:10.3389/fvets.2020.00201.
  4. Zhuang, C.; Liu, G.; Barkema, H.W.; Zhou, M.; Xu, S.; Ur Rahman, S.; Liu, Y.; Kastelic, J.P.; Gao, J.; Han, B. Selenomethionine Suppressed TLR4/NF-κB Pathway by Activating Selenoprotein S to Alleviate ESBL Escherichia coli-Induced Inflammation in Bovine Mammary Epithelial Cells and Macrophages. Frontiers in microbiology 2020, 11, 1461, doi:10.3389/fmicb.2020.01461.
  5. Xue, Y.; Du, M.; Zhu, M.-J. Quercetin suppresses NLRP3 inflammasome activation in epithelial cells triggered by Escherichia coli O157:H7. Free Radical Biology and Medicine 2017, 108, 760-769, doi:10.1016/j.freeradbiomed.2017.05.003.
  6. Xue, Y.; Du, M.; Sheng, H.; Hovde, C.J.; Zhu, M.J. Escherichia coli O157:H7 suppresses host autophagy and promotes epithelial adhesion via Tir-mediated and cAMP-independent activation of protein kinase A. Cell Death Discov 2017, 3, 17055, doi:10.1038/cddiscovery.2017.55.
  7. Yu, L.C.; Wei, S.C.; Li, Y.H.; Lin, P.Y.; Chang, X.Y.; Weng, J.P.; Shue, Y.W.; Lai, L.C.; Wang, J.T.; Jeng, Y.M.; et al. Invasive Pathobionts Contribute to Colon Cancer Initiation by Counterbalancing Epithelial Antimicrobial Responses. Cell Mol Gastroenterol Hepatol 2022, 13, 57-79, doi:10.1016/j.jcmgh.2021.08.007.
  8. Neubert, P.; Weichselbaum, A.; Reitinger, C.; Schatz, V.; Schröder, A.; Ferdinand, J.R.; Simon, M.; Bär, A.L.; Brochhausen, C.; Gerlach, R.G.; et al. HIF1A and NFAT5 coordinate Na(+)-boosted antibacterial defense via enhanced autophagy and autolysosomal targeting. Autophagy 2019, 15, 1899-1916, doi:10.1080/15548627.2019.1596483.
  9. Li, Q.; Liao, J.; Chen, W.; Zhang, K.; Li, H.; Ma, F.; Zhang, H.; Han, Q.; Guo, J.; Li, Y.; et al. NAC alleviative ferroptosis in diabetic nephropathy via maintaining mitochondrial redox homeostasis through activating SIRT3-SOD2/Gpx4 pathway. Free radical biology & medicine 2022, 187, 158-170, doi:10.1016/j.freeradbiomed.2022.05.024.
  10. Wang, T.; Ye, X.; Bian, W.; Chen, Z.; Du, J.; Li, M.; Zhou, P.; Cui, H.; Ding, Y.Q.; Qi, S.; et al. Allopregnanolone Modulates GABAAR-Dependent CaMKIIδ3 and BDNF to Protect SH-SY5Y Cells Against 6-OHDA-Induced Damage. Frontiers in cellular neuroscience 2019, 13, 569, doi:10.3389/fncel.2019.00569.
  11. Pan, X.; Li, R.; Guo, H.; Zhang, W.; Xu, X.; Chen, X.; Ding, L. Dihydropyridine Calcium Channel Blockers Suppress the Transcription of PD-L1 by Inhibiting the Activation of STAT1. Frontiers in pharmacology 2020, 11, 539261, doi:10.3389/fphar.2020.539261.
  12. Zhan, C.S.; Chen, J.; Chen, J.; Zhang, L.G.; Liu, Y.; Du, H.X.; Wang, H.; Zheng, M.J.; Yu, Z.Q.; Chen, X.G.; et al. CaMK4-dependent phosphorylation of Akt/mTOR underlies Th17 excessive activation in experimental autoimmune prostatitis. FASEB journal : official publication of the Federation of American Societies for Experimental Biology 2020, 34, 14006-14023, doi:10.1096/fj.201902910RRR.

Round 2

Reviewer 1 Report

: However it is only speculation, it is not valued to read.

Author Response

Comments: However, it is only speculation, it is not valued to read.

Point: Thank you for your comments and wish you a happy life.

Reviewer 3 Report

I have no further questions. The paper is of a sufficient level to be published and I recommend its acceptance.

Author Response

Comments and Suggestions for Authors:I have no further questions. The paper is of a sufficient level to be published and I recommend its acceptance.

Point: Thank you for your comments and wish you a happy life.